# Impact of Operator Medical Specialty on Endotracheal Intubation Rates in Prehospital Emergency Medicine—A Retrospective Cohort Study

**DOI:** 10.3390/jcm11071992

**Published:** 2022-04-02

**Authors:** Christophe A. Fehlmann, Michèle Chan, Romain Betend, Fiona Novotny-Court, Mélanie Suppan, Georges L. Savoldelli, Laurent Suppan

**Affiliations:** 1Division of Emergency, Department of Anaesthesiology, Clinical Pharmacology, Intensive Care and Emergency Medicine, Geneva University Hospitals, Faculty of Medicine, University of Geneva, 1211 Geneva, Switzerland; michele.chan@hcuge.ch (M.C.); romain.betend@hcuge.ch (R.B.); laurent.suppan@hcuge.ch (L.S.); 2School of Epidemiology and Public Health, University of Ottawa, Ottawa, ON K1G 5Z3, Canada; 3Ottawa Hospital Research Institute, Ottawa, ON K1Y 4E9, Canada; 4Division of Internal and General Medicine, Department of Medicine, Geneva University Hospitals, Faculty of Medicine, University of Geneva, 1211 Geneva, Switzerland; fiona.novotny@hcuge.ch; 5Division of Anaesthesiology, Department of Anaesthesiology, Clinical Pharmacology, Intensive Care and Emergency Medicine, Geneva University Hospitals, Faculty of Medicine, University of Geneva, 1211 Geneva, Switzerland; melanie.suppan@hcuge.ch (M.S.); georges.savoldelli@hcuge.ch (G.L.S.); 6Unit of Development and Research in Medical Education (UDREM), Faculty of Medicine, University of Geneva, 1211 Geneva, Switzerland

**Keywords:** intubation, prehospital, specialty, anaesthesiologist, supervision, education, training

## Abstract

Prehospital endotracheal intubation (ETI) can be challenging, and the risk of complications is higher than in the operating room. The goal of this study was to compare prehospital ETI rates between anaesthesiologists and non-anaesthesiologists. This retrospective cohort study compared prehospital interventions performed by either physicians from the anaesthesiology department (ADP) or physicians from another department (NADP, for non-anaesthesiology department physicians). The primary outcome was the prehospital ETI rate. Overall, 42,190 interventions were included in the analysis, of whom 68.5% were performed by NADP. Intubation was attempted on 2797 (6.6%) patients, without any difference between NADPs and ADPs (6.5 versus 6.7%, *p* = 0.555). However, ADPs were more likely to proceed to an intubation when patients were not in cardiac arrest (3.4 versus 3.0%, *p* = 0.026), whereas no difference was found regarding cardiac arrest patients (65.2 versus 67.7%, *p* = 0.243) (*p* for homogeneity = 0.005). In a prehospital physician-staffed emergency medical service, overall ETI rates did not depend on the frontline operator’s medical specialty background. ADPs were, however, more likely to proceed with ETI than NADPs when patients were not in cardiac arrest. Further studies should help to understand the reasons for this difference.

## 1. Introduction

In the prehospital setting, airway management is often required in critical situations, such as cardiac arrest [1,2,3], respiratory distress, coma, or traumatic injuries [4,5]. In such situations, endotracheal intubation (ETI) is the most effective technique to secure the airways [6]. Difficult and sometimes even hostile field conditions, restricted ergonomics, limited equipment and human resources, and patients’ critical clinical conditions all make prehospital ETI particularly challenging [7,8].

Prehospital emergency medical systems (EMSs) differ considerably from one country or region to another [9]. Depending on the system, ETIs can be performed by different types of providers, some of whom can be rather inexperienced regarding airway management procedures [10,11]. In recent studies, prehospital ETI success rates ranging from 58 to 100% have been reported [10,12,13,14]. In physician-staffed EMSs, background, medical specialty, experience, and training all affect ETI success rates and ETI-related outcomes [15,16,17,18].

The medical background of prehospital physicians is subject to considerable variability. Anaesthesiologists, who are most familiar with airway management techniques, are often present in physician-staffed EMSs [13,19]. Nevertheless, general practitioners and other specialists, such as surgeons and even radiologists, also operate in some settings [17,20]. Depending on their curriculum, senior emergency physicians can also be proficient in airway management [15,17], but emergency medicine residents often lack competence in advanced airway management. In addition, the level of expertise of prehospital physicians varies significantly as some systems employ junior residents while others only recruit certified senior specialists [14,18,21].

Physician-staffed EMSs have higher ETI success rates than systems that only employ paramedics, with the highest success rates achieved when experienced specialists either directly intervened or supervised prehospital interventions [10,11,22,23]. Moreover, complications and unfavourable ETI-related outcomes occur less often in physician-staffed systems [11]. Among the many reasons underlying these results, the need to perform an anaesthetic induction, that is, administration of a hypnotic drug, is paramount. Indeed, while most cardiac arrest patients can be intubated without using specific drugs, ETI is facilitated in most other patients by the administration of a hypnotic drug and a neuromuscular blocking agent [10,24]. As the use of such drugs is associated with many untoward side effects, such as arrhythmia, aspiration, oxygen desaturation, or hypotension, many prehospital medical directors are reluctant to allow providers other than physicians to use them in the field [25]. Indeed, extensive knowledge and training are required to safely and adequately use hypnotics and neuromuscular blocking agents, and physicians are expected to take into account each patient’s condition and specificities and to adapt their therapeutic strategy accordingly. Therefore, physicians belonging to specific medical specialties, such as anaesthesiologists, might be more attentive to the adverse effects of such drugs and more wary of potential ETI-related complications. These physicians might be more reluctant to perform prehospital ETIs even though they practice this procedure more frequently than non-anaesthesiologists [15]. One could, however, also argue that anaesthesiologists are also more comfortable with airway management procedures and more wary of adverse events linked to an altered level of consciousness, such as aspiration. Therefore, they might resort to ETI with less hesitation than non-anaesthesiologists. Consequently, assessing the existence of an actual difference in ETI rates according to medical specialty is scientifically relevant.

The aim of this study was to evaluate ETI rates in a resident-staffed and specialist physician-supervised EMS [14] according to the operator’s medical specialty background (anaesthesiologist vs. non-anaesthesiologist).

## 2. Materials and Methods

### 2.1. Study Design and Setting

This study’s design was an observational retrospective cohort study and its report follows the STrengthening the Reporting of OBservational studies in Epidemiology (STROBE) Statement guidelines [26]. It is based on data related to interventions performed by the prehospital unit of Geneva University Hospital, the structure of which has been described in prior publications [14]. A previous paper, purely descriptive, was published using the same dataset [14]. Briefly, the prehospital EMS operating in Geneva is three-tiered. The first tier is made of advanced life support ambulances staffed by paramedics, who are not allowed to intubate, while the second tier is composed of a medical mobile unit (called SMUR for Service Mobile d’Urgence et de Réanimation), which is operated by an advanced paramedic and by a frontline physician. Most of these physicians are senior residents who belong to one of three different services: anaesthesiology, emergency medicine, or internal medicine [27]. Their level of experience and ETI skills are therefore heterogeneous, and a senior specialist physician provides distance or on-site supervision round the clock [14]. At the beginning of their prehospital rotation, which usually lasts between three and four months, every physician receives a formal training regarding prehospital intubation, independently of their specialty. Then, during their rotation, frontline physicians who are not considered to have a sufficient background in anaesthesiology are systematically supervised on site whenever an anaesthetic induction has to be performed to facilitate ETI. Once deemed proficient enough, frontline physicians are allowed to intubate all patients in whom no ETI difficulty is anticipated regardless of their service of origin. During the study period, laryngoscopies were carried out using either direct laryngoscopes or Airtraqs (Prodol Meditec, Vizcaya, Spain). The kind of laryngoscope used was not recorded in the medical files.

After each intervention, a comprehensive computerised medical file is filled by the frontline physician. All records are reviewed daily by a supervisor for teaching and quality control purposes. Data is stored in the institutional database.

The computerised medical files of all adult patients (18 years or older) taken care of by an SMUR unit between 1 January 2008 and 31 December 2018 were included. We then excluded files reporting distance (phone) consultations, major incidents with multiple casualties, interventions performed by a supervisor acting as a frontline physician, and intervention with patients uninjured or not found. These latter interventions were excluded because supervisors are only dispatched to take care of the most critical situations when a frontline SMUR unit is unavailable. Finally, interventions were excluded when the physician in charge could not be properly identified or when data regarding their curriculum or medical background could not be retrieved.

### 2.2. Variables and Outcomes

The main predictor variable was the medical specialty to which the physician belonged. We defined two groups: physicians coming from the anaesthesiology department (ADP, for anaesthesiology department physicians) and physicians coming from another department (emergency medicine or internal medicine) (NADP, for non-anaesthesiology department physicians).

The primary outcome was the rate of ETI attempted by frontline physicians, regardless of ETI success. Secondary outcomes included the overall ETI success rate and ETI success at first attempt, where success was defined as capnographic confirmation of the correct placement of the endotracheal tube [28]; number of ETI attempts, defined as insertion of the laryngoscope blade into the oropharynx whether or not an attempt to insert the endotracheal tube was made [29]; and intubation supervision rate.

All intervention-related variables, including ETI data, patient’s age and sex, place of intervention, time of intervention (night, weekend), primary pathology (cardiac arrest versus non-cardiac arrest), and patient severity (assessed using the National Advisory Committee for Aeronautics—NACA—score [30]), were automatically extracted from the institutional database.

Data regarding physicians were collected from the SMUR schedule archives and additional information was provided by the human resources department.

### 2.3. Statistical Analysis

The baseline characteristics of the patients are presented as a mean with standard deviation and frequency with proportion. We then compared interventions between both groups (ADP versus NADP), using Student’s t test and chi-squared test as appropriate. Normality was assumed based on the high number of patients included in our study.

For the primary outcome, we reported the frequency and proportion, compared both groups using the chi-squared test, and reported the risk ratio. We did not perform any multivariable analyses because the exposure (ADP or NADP) was based on a random allocation (depending purely on the schedule and the availability of the unit). Two pre-specified subgroup analyses were performed. First, we compared patients who were in cardiac arrest before intubation to those who were not. Second, we looked for a potential effect modification according to the sex of the physician. Similar methods were used to analyse secondary outcomes. Additionally, we used a test to assess the trend in the number of ETI attempts.

Finally, the null hypothesis was that there would be no difference in the ETI rates between the ADP and NADP groups. Assuming an overall intubation rate of 5% and a NADP to ADP ratio of 2:1, we estimated that approximately 22,000 patients would be required to have a 90% chance of detecting, at the 5% significance level, a relative difference of 20% in the primary outcome measure. No imputation was performed, and missing data were coded and analysed accordingly. A 2-sided *p* value below 0.05 was considered significant. Statistical analysis was performed using Stata 16 (StataCorp, College Station, TX, USA).

### 2.4. Ethics Approval

This study was approved on 23 April 2019, by the institutional ethics committee of Geneva, Switzerland (Project ID 2019-00679, Chairperson Pr Bernard Hirschel). The patient consent was waived by the committee. Moreover, all experiments were performed in accordance with relevant guidelines and regulations.

## 3. Results

Between January 2008 and December 2018, 53,721 files were filled for interventions for adults (Figure 1). We excluded 11,531 interventions, mainly because they were only phone consultations for paramedics, or were performed by a supervisor alone. Of the remaining 42,190 interventions, 13,274 (31.5%) were performed by ADPs and 28,916 (68.5%) by NADPs. Male physicians performed 24,073 interventions (57.1%), with a higher proportion in the NADP group than in the ADP group (59.6 vs. 51.5%, *p* < 0.001).

Table 1 reports the patients’ characteristics. Patients had a mean age of 62.4 years (SD = 21.1), and most of them were male (54.9%). Interventions most often took place at home (61.4%) and during the day (60.5%). There were not any clinically relevant differences between interventions performed by ADPs and NADPs.

During the research period, intubation was attempted in 2797 (6.6%) patients, without any difference between NADPs and ADPs (6.6 vs. 6.7%, *p* = 0.555) (Table 2). There was no modification effect by physician sex (*p* for homogeneity = 0.070). However, ADPs were more likely to proceed to an intubation when patients were not in cardiac arrest (3.4 vs. 3.0%, *p* = 0.026), whereas no difference was found in cardiac arrest patients (65.2 vs. 67.7%, *p* = 0.243) (*p* for homogeneity = 0.005) (Table 3).

Among the patients in whom ETI was attempted, the overall success rate was 96.6%, without any difference between NADPs and ADPs (96.3 versus 97.2%, *p* = 0.206) (Table 2). The success rate at first attempt was 72.1%, with an important difference between NADPs and ADPs (67.5 vs. 81.9%, *p* < 0.001). Regarding the number of attempts, there was also a strong correlation with the medical department the physician belonged to (*p* for trend < 0.001). Finally, NADPs performing an intubation were more likely to be supervised (either by phone or on-site) than ADPs (*p* < 0.001). While the ETI success rate was globally lower in case of cardiac arrest, no modification effect was found for the association between department and secondary outcomes in this subgroup (Table 3).

## 4. Discussion

This study shows that, in a supervised prehospital physician-staffed EMS, the decision to attempt ETI depends little on the physician’s specialty. There was no difference when only considering patients in cardiac arrest while ADPs were more likely to intubate patients for whom the use of hypnotic and neuromuscular blocking agents was required. While overall success was similar between both groups, ADPs had higher ETI success rates at the first attempt, needed fewer attempts to succeed, and were less often supervised.

Different hypotheses can be formulated regarding the primary outcome and modification effect by cardiac arrest. First, some physicians might be reluctant to call upon their supervisor to help them perform ETI in unclear situations. As NADPs cannot perform ETI without being supervised during the first weeks or months of their prehospital rotation, they might choose to transport rather than call for help when the clinical situation does not clearly require advanced airway management procedures. Similarly, supervisors can, at times, already be dealing with another mission and could, therefore, be unavailable to assist frontline physicians. According to their experience, these physicians could decide to transport the patient without proceeding with ETI rather than wait for the supervisor to become available. Finally, it is possible that their background helps ADPs better anticipate airway patency issues. This could lead them to consider broader ETI indications than NADPs. Two contradictory hypotheses can be formulated in this regard. Indeed, the first implies that ADPs were warier of potential airway complications than NADPs, and decided to move on to ETI early and adequately. Conversely, ADPs might also have felt overconfident and could have taken unnecessary risks by performing debatable prehospital ETIs. Further studies are required to elucidate these findings.

ADPs had much higher rates of success at first attempt and needed fewer attempts. This is in line with previous studies showing that non-anaesthesiologists are less successful at performing ETI, especially when rapid sequent induction is required [16,17]. In some settings, multiple ETI attempts have been associated with complications and with worse outcomes [31]. Unfortunately, data regarding these safety issues was not available in the prehospital medical files and we were therefore unable to determine the actual consequences of this difference. It is, however, probable that the lower ETI success rate at first attempt found when ETI was performed by NAPDs is linked to the willingness of most supervisors to teach them the procedure. In this context, complications might occur less often even with a higher number of ETI attempts by virtue of supervision. The high overall ETI success rate, which was identical between groups, probably also reflects the advantages conferred by senior supervision in this physician-staffed system [14]. The ETI success rate at first attempt was, however, lower than that reported in previous papers describing similar intubation rates [32,33]. This is easily explained by the fact that, in our system, providers are in training and have less ETI experience than the providers who operated in Gellerfors’ cohort [33].

Some strengths and limitations should be acknowledged. First, while several studies have examined ETI-related outcomes according to physician specialty, this might be the first to report ETI rate by specialty, thus adding to current knowledge. In addition, the important sample size and quality of the data should be considered. With more than 40,000 interventions, this study was sufficiently powered to demonstrate a potential effect and to study different interactions. The quality of the data also reinforces the internal validity of the results, as all prehospital medical files are validated by supervisors and there are very few missing values. On the other hand, internal validity could be jeopardised by the independent variable we used. While physicians sent by the anaesthesiology department obviously have some training regarding airway management procedures, this training could be quite different between a young resident and a more experienced physician. While adjusting for the number of previous ETI would have provided additional information, both this variable and the number of months spent in anaesthesiology training could not be retrieved. Moreover, some physicians working in the ED could have undergone one year of anaesthesiology training during their curriculum and would nevertheless have been considered as NAPDs in the present study. However, in our setting, only a few physicians would have followed this course, and this should not have a significant effect on our results. Even though most ETIs were carried out in patients who were in cardiac arrest, it would have been interesting to further categorize the reasons for performing ETI in other patients. We were, however, unable to extract this data. Finally, the specificity of our system could prevent the generalisation of our results to other EMSs.

The difference in ETI rates found in patients who were not in cardiac arrest should place our local training and supervision under question, as patient management should not be influenced by the profile of the physician in charge. As our study was not designed to determine whether ETIs were actually and uncontrovertibly indicated in either group, the next step should be to identify the characteristics of patients who were intubated by ADPs but not by NADPs to refine airway management guidelines. The potential complications linked to multiple ETI attempts in the setting of this supervised physician-staffed system should also be assessed.

## 5. Conclusions

In a prehospital physician-staffed EMS, the overall ETI rates did not depend on the medical specialty of the frontline physician. However, ADPs were more likely to proceed with ETI than NADPs when patients were not in cardiac arrest, probably owing to the need to use hypnotic and neuromuscular blocking agents and had higher ETI success rates at first attempt. Further studies are needed to determine the consistency and adequacy of the indications for ETI.

## Figures and Tables

**Figure 1 jcm-11-01992-f001:**
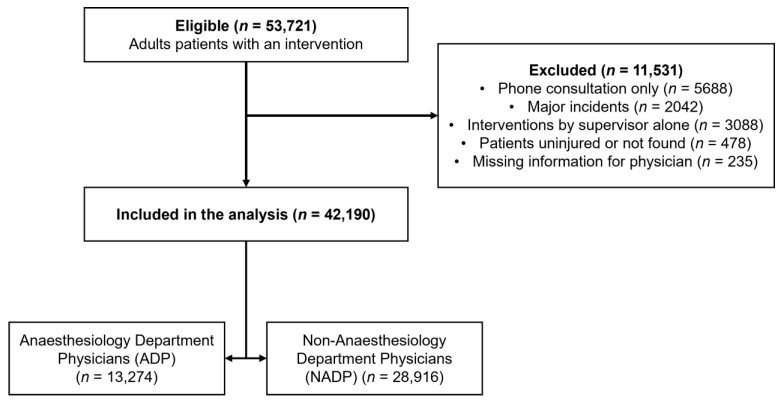
Flowchart of prehospital interventions’ inclusion.

**Table 1 jcm-11-01992-t001:** Patients’ characteristics.

	Overall (N = 42,190)	Non-Anaesthesiology Department Physicians (N = 28,916)	Anaesthesiology Department Physicians (N = 13,274)	*p*-Value
Male—*n* (%)	23,426 (54.9)	16,084 (55.0)	7342 (54.7)	0.585
Age (years)—mean ± SD	62.4 ± 21.1	62.3 ± 21.1	62.7 ± 21.0	0.093
Intervention site—*n* (%)				0.847
Home	25,916 (61.4)	17,749 (61.4)	8167 (61.5)	
Public place	9705 (23.0)	6674 (23.1)	3031 (22.8)	
Healthcare place	6569 (15.6)	4493 (15.5)	2076 (15.6)	
Week-end—*n* (%)	11,082 (26.3)	7607 (26.3)	3475 (26.2)	0.781
Night—*n* (%)	16,680 (39.5)	11,206 (38.8)	5474 (41.2)	<0.001
Heart rate (/min)—mean ± SD	93.5 ± 28.0	93.7 ± 28.1	92.9 ± 27.8	0.005
Systolic blood pressure (mmHg)—mean ± SD	138.6 ± 31.6	138.7 ± 31.7	138.4 ± 31.6	0.478
Respiratory rate (/min)—mean ± SD	21.9 ± 8.4	21.9 ± 8.4	22.0 ± 8.3	0.191
Oxygen saturation (%)—mean ± SD	94.9 ± 7.4	94.9 ± 7.4	94.8 ± 7.2	0.630
Glasgow coma scale—*n* (%)				<0.001
<8	3963 (9.4)	2814 (9.7)	1149 (8.7)	
>7	26,388 (62.6)	18,116 (62.7)	8272 (62.3)	
*Missing*	11,839 (28.1)	7986 (27.6)	3853 (29.0)	
Cardiac arrest—*n* (%)				0.492
No	39,986 (94.5)	27,322 (94.5)	12,564(94.7)	
Yes	2304 (5.3)	1594 (5.5)	710 (5.3)	
NACA—*n* (%)				0.146
1-2-3	12,329 (29.2)	8393 (29.0)	3396 (29.7)	
4-5-6-7	29,784 (70.6)	20,476 (70.8)	9308 (70.1)	
*Missing*	77 (0.2)	47 (0.2)	30 (0.2)	

**Table 2 jcm-11-01992-t002:** Outcomes.

	Overall (N = 42,190)	Non-Anaesthesiology Department Physicians (N = 28,916)	Anaesthesiology Department Physicians (N = 13,274)	*p*-Value
1° Outcome				
Intubation—*n* (%)	2797 (6.6)	1903 (6.6)	894 (6.7)	0.555
2° Outcomes *				
Success—*n* (%)	2701 (96.6)	1832 (96.3)	869 (97.2)	0.206
Success at first attempt—*n* (%)	2016 (72.1)	1284 (67.5)	732 (81.9)	<0.001
Intubation attempts—*n* (%)				<0.001
1	2016 (72.1)	1284 (67.5)	732 (81.9)	
2	574 (20.5)	445 (28.4)	129 (14.4)	
3	158 (5.7)	136 (7.2)	22 (2.5)	
>3	39 (1.4)	30 (1.6)	9 (1.0)	
*Missing*	10 (0.4)	8 (0.4)	2 (0.2)	
Supervision—*n* (%)				<0.001
None	1248 (44.6)	730 (38.4)	518 (57.9)	
By phone	429 (15.3)	285 (15.0)	144 (16.1)	
On site	1120 (40.0)	888 (46.7)	232 (26.0)	

*: percentages are among patients with the primary outcome.

**Table 3 jcm-11-01992-t003:** Modification effect by cardiac arrest status for selected outcomes.

	No Cardiac Arrest (N = 39,886)		In Cardiac Arrest (2304)		
	Non-Anaesthesiology Department Physicians (N = 27,322)	Anaesthesiology Department Physicians (N = 12,564)	Non-Anaesthesiology Department Physicians (N = 1594)	Anaesthesiology Department Physicians (N = 710)	*p*-Value ^#^
1° Outcome					
Intubation—*n* (%)	824 (3.0)	431 (3.43)	1079 (67.7)	463 (65.2)	0.005
2° Outcomes *					
Success—*n* (%)	815 (98.9)	427 (99.1)	1017 (94.3)	442 (95.5)	0.287
Success at first attempt—*n* (%)	581 (70.5)	366 (84.9)	703 (65.2)	366 (79.1)	0.868

*: percentages are among patients in whom intubation was attempted; ^#^: *p*-values are for the test of homogeneity using the Cochran–Mantel–Haenszel method.

## Data Availability

The data that support the findings of this study are freely available on the Open Science Framework (https://doi.org/10.17605/OSF.IO/HZSX3, accessed on 13 February 2022).

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
