# Peer review of "Impact of Operator Medical Specialty on Endotracheal Intubation Rates in Prehospital Emergency Medicine—A Retrospective Cohort Study"

_jcm, 2022, doi:10.3390/jcm11071992_

Round 1

Reviewer 1 Report

Dear editor and authors, thanks for the opportunity to review this manuscript titled :  Impact of operator medical specialty on endotracheal intubation rates in prehospital emergency medicine – A retrospective 
cohort study. 

The study aims to review the rate of prehospital ETI rates 
between anaesthesiologist versus non-anaesthesiologist. Overall  they found no differences between the two groups and even though the anaesthesiologists were prone to intubate more when patients were not in cardiac  arrest.

The study is well written and it describes a specific setting of  prehospital emergency medical systems  in Geneva which shows the difference of practice. 

Abstract: please describe EMS in the abstract as for the other abbreviations.

Discussion: last paragraph please change cardiac into cardiac arrest.

In the same paragraph It should be also prioritize the  impact of this study on the actual setting and training of the supervised physician-staffed system in place and the  potential complications linked 
to multiple ETI attempts in the setting of this supervised physician-staffed system. This because this outcomes were not looked into in the results.

Author Response

Thank you very much for the review. Please find our reply attached.

Reviewer 1

Comments from reviewer

Answers

Modification in text

Localisation

The study is well written and it describes a specific setting of  prehospital emergency medical systems  in Geneva which shows the difference of practice. 

Thank you for this comment regarding our manuscript.

Abstract: please describe EMS in the abstract as for the other abbreviations.

Thank you very much for pointing this out. It was modified in the text.

p1

Discussion: last paragraph please change cardiac into cardiac arrest.

Thank you again for pointing out this typo. It was modified in the text.

p7

In the same paragraph It should be also prioritize the  impact of this study on the actual setting and training of the supervised physician-staffed system in place and the  potential complications linked 
to multiple ETI attempts in the setting of this supervised physician-staffed system. This because this outcomes were not looked into in the results

You are perfectly right: even there might be a risk related to multiple ETI attempts, this risk could be mitigated by virtue of supervision, but our study design prevented us from assessing it. Therefore, we modified the text.

The specificity of our system was already acknowledged at the end of the discussion (“Finally, the specificity of our system could prevent the generalisation of our results to other EMSs”)

In this context, complications might occur less often even with a higher number of ETI attempts by virtue of supervision.

p7

Reviewer 2 Report

Overall, the authors describe an observational study looking at out-of-hospital intubation and comparison between anesthesia versus non-anesthesia physicians. They also describe subgroup analysis in cardiac arrest and non-cardiac arrest patient. While the study design, hypothesis and manuscript is well written, below are minor recommendations from this reviewer:

  1. Were all intubations performed by physicians? Were some intubations performed by paramedics
  2. For non-cardiac arrest intubation, it would be meaningful to explain the reason for intubation in table 1/descriptive statistics as rate of intubation was different in each cohort. 
  3. Since such large dataset is being used, was normality assumption fulfilled for continuous variables. If so, then student T-test is appropriate, if not, then non-parametric tests such as Mann-Whitney Test should be utilized for continuous variables. Please ensure, appropriate biostatistical methodology is utilized.
  4. Chi-squared test is appropriate for categorical variables, please replace the term "Chi-2" to "chi-squared".
  5. In the last paragraph of the manuscript, "The difference in ETI rates found in patients who were not in cardiac "ARREST" should question 
    our local training and supervision": word "arrest" is missing.
  6. Otherwise, no major issues.

Author Response

Thank you very much for the review. Please find our reply attached.

Reviewer 2

Comments from reviewer

Answers

Modification in text

Localisation

While the study design, hypothesis and manuscript is well written, below are minor recommendations from this reviewer:

We would like to thank you for the time spent to review our manuscript and your positive comments regarding the design and the writing.

Were all intubations performed by physicians? Were some intubations performed by paramedics

Yes, all the intubations are performed by physicians. Paramedics are not allowed to intubate in our system. This is now clearly acknowledged in the methods.

The first tier is made of advanced life support ambulances staffed by paramedics, who are not allowed to intubate, while the second tier is composed of a medical mobile unit (called SMUR for Service Mobile d’Urgence et de Réanimation) which is operated by an advanced paramedic and by a frontline physician.

p3

For non-cardiac arrest intubation, it would be meaningful to explain the reason for intubation in table 1/descriptive statistics as rate of intubation was different in each cohort.

Your comment is extremely relevant. However, this data is not routinely collected, and it would imply to review each case to look at the reason of the intubation. Therefore, given the number of intubations, the important workload, and the potential errors linked to this manual review, we were unable to address this issue. In line with your comment and with this shortcoming, the following sentence has been added to the limitations section.

Even though most ETIs were carried out in patients who were in cardiac arrest, it would have been interesting to further categorize the reasons for performing ETI in other patients. We were however unable to extract this data.

p7

Since such large dataset is being used, was normality assumption fulfilled for continuous variables. If so, then student T-test is appropriate, if not, then non-parametric tests such as Mann-Whitney Test should be utilized for continuous variables. Please ensure, appropriate biostatistical methodology is utilized.

You are perfectly right. We considered that, based on the large number of patients, the central limit theorem could be applied, and normality can therefore be assumed. We added a sentence in the method.

Normality was assumed based on the high number of patients included in our study.

p4

Chi-squared test is appropriate for categorical variables, please replace the term "Chi-2" to "chi-squared".

Practice can be different from a journal to another. We did not see any guidelines regarding this for this journal. We changed it following your comment, but if the editor knows a standard for MDPI journal we would be happy to conform to it.

p4

In the last paragraph of the manuscript, "The difference in ETI rates found in patients who were not in cardiac "ARREST" should question our local training and supervision": word "arrest" is missing.

Thank you for pointing out this typo. It was modified in the text.

p7